# Disruption of awake sharp-wave ripples does not affect memorization of locations in repeated-acquisition spatial memory tasks

**Lies Deceuninck[1,2]\*, Fabian Kloosterman[2,3]\***

[1]KU Leuven, Department of Physics and Astronomy, Soft Matter and Biophysics, Heverlee, Belgium; [2]NERF-NeuroElectronics Research Flanders, Kloosterman Lab, Heverlee, Belgium; [3]KU Leuven, Faculty of Psychology & Educational Sciences, Leuven, Belgium

**Abstract** Storing and accessing memories is required to successfully perform day-to-day tasks, for example for engaging in a meaningful conversation. Previous studies in both rodents and primates have correlated hippocampal cellular activity with behavioral expression of memory. A key role has been attributed to awake hippocampal replay – a sequential reactivation of neurons representing a trajectory through space. However, it is unclear if awake replay impacts immediate future behavior, gradually creates and stabilizes long-term memories over a long period of time (hours and longer), or enables the temporary memorization of relevant events at an intermediate time scale (seconds to minutes). In this study, we aimed to address the uncertainty around the time-frame of impact of awake replay by collecting causal evidence from behaving rats. We detected and disrupted sharp wave ripples (SWRs) - signatures of putative replay events - using electrical stimulation of the ventral hippocampal commissure in rats that were trained on three different spatial memory tasks. In each task, rats were required to memorize a new set of locations in each trial or each daily session. Interestingly, the rats performed equally well with or without SWR disruptions. These data suggest that awake SWRs - and potentially replay - does not affect the immediate behavior nor the temporary memorization of relevant events at a short timescale that are required to successfully perform the spatial tasks. Based on these results, we hypothesize that the impact of awake replay on memory and behavior is long-term and cumulative over time.

**\*For correspondence:**
lies.deceuninck@gmail.com (LD);
kloosterman.fabian@gmail.com
(FK)

**Competing interest:** The authors declare that no competing interests exist.

## Editor's evaluation

This study reports lack of effect of closed-loop disruption of awake sharp-wave ripples in the repeated-acquisition of spatial memory tasks. These negative results have important theoretical and practical implications in the field of learning and memory. The strength of evidence is solid with methods, data, and analyses broadly supporting the claims.

## Introduction

In optimizing our behavior, we benefit from our past experiences. For this, storing and accessing the appropriate memory traces at the right time is crucial and it is believed that the hippocampus is engaged in both of these processes. Of particular interest are hippocampal sequence reactivation (replay) events in the awake state (*Foster and Wilson, 2006*; *Diba and Buzsáki, 2007*). During replay, hippocampal place cell spike sequences, that represent a trajectory through space, are briefly

re-activated during quiescent episodes and engage a wide network of brain regions. It has been hypothesized that these neuronal reactivations contribute to memory both in humans (e.g. *Huang et al., 2018*; *Tambini and Davachi, 2013*; *Staresina et al., 2013*; *Liu et al., 2019*; *Norman et al., 2019*) and rodents (e.g. *Singer et al., 2013*; *Pfeiffer and Foster, 2013*; *Xu et al., 2019*).

However, it is unclear over what timeframe the hippocampally driven reactivations impact memory-guided behavior. One possibility is that reactivation is pertinent for immediate future behavior, as is the case in decision-making. For example, replay sequences could reflect potential upcoming trajectories and contribute to making an optimized navigational choice based on past experiences. A second possibility is that awake replay has only minimal immediate impact on memory-guided behavior, but rather has a gradual and cumulative effect over a long period of time (hours and longer) on the creation and stabilization of long-term memories. This is generally the idea behind the role of sleep replay in memory consolidation, which is thought to slowly reorganize cortical networks and reduce the dependency of new memories on the hippocampus (*Squire et al., 2015*). Finally, a third possibility is that awake replay enables the temporary memorization of relevant events at an intermediate time scale. In this case, the events could serve to fill or refresh a temporary storage buffer for events that need to be remembered over a short time span (seconds to minutes).

Evidence from studies that correlate awake replay to behavior can be interpreted to support each of the timeframes - immediate, intermediate, and long-term. Several reports indicate that replay content appears to reflect trajectories to follow (*Pfeiffer and Foster, 2013*; *Xu et al., 2019*) or avoid (*Wu et al., 2017*; *Gupta et al., 2010*; *Carey et al., 2019*), or may reflect different navigation options (*Singer et al., 2013*; *Ólafsdóttir et al., 2017*; *Shin et al., 2019*; *Gillespie et al., 2021*). These data are consistent with a role of awake replay in guiding the upcoming navigation choices. Other reports indicate that awake replay reflects recently experienced trajectories. These replay events most often occur after the receipt of a reward (*Singer and Frank, 2009*) and often represent trajectories toward rewarded locations (*Ambrose et al., 2016*; *Diba and Buzsáki, 2007*; *Foster and Wilson, 2006*; *Karlsson and Frank, 2009*) or even represent an optimized trajectory before it has reliably been taken by the animal (*Igata et al., 2021*). In these cases, awake replay appears to be biased toward useful past experiences that are not immediately relevant but might need to be remembered for intermediate or long-term to optimize future behavior.

While correlative studies could provide clues on what drives the expression of awake replay, they cannot answer the question in what way and over what timeframe awake replay supports memory-guided behavior. To address this question, causal evidence based on selective manipulation of replay events is needed. As a first approach, previous studies have looked at awake sharp-wave ripples (SWR), which are short (30–100ms) high-frequency (160–225 Hertz) oscillations in the local field potential (LFP) in the hippocampus (*Buzsáki et al., 1992*). Hippocampal replay is strongly associated with the occurrence of SWRs and replay events are often considered to be a subset of SWR events. The characteristic oscillatory pattern of SWRs makes is possible to detect and manipulate them in real time (*Ciliberti and Kloosterman, 2017*).

When awake SWRs are disrupted or prolonged, rats learn a spatial alternation task significantly slower (*Jadhav et al., 2012*; *Igata et al., 2021*) or faster (*Fernández-Ruiz et al., 2019*), respectively. A common interpretation of these experiments is that SWRs (and by extension replay) support spatial 'working' memory, that is, memorizing the most recently visited location to select a subsequent path. However, given that in these experiments, the rats learned a stereotypical pattern of maze arm visits across several days, other explanations for the role of SWRs cannot be ruled out, including memorizing the temporal order of events and supporting long-term consolidation. Overall, the existing causal studies have not unequivocally demonstrated whether SWRs (or replay) impact the learning process and behavior at the immediate, intermediate, or long-term timescales.

This paper aims to narrow down the timeframe over which awake replay impacts memory-based behavior by performing SWR disruption in tasks with memory demands over a short and intermediate timeframe. We report three separate experiments in which the performance of rats on a repeated-acquisition spatial memory task is not influenced by disruption of awake SWR. Crucially, in all these behavioral tasks, rats were required to memorize a new set of trial-specific or session-specific locations. High performance on the tasks could not be obtained by relying on long-term (e.g. from the previous days) to achieve a high performance, but required memories of recent events and the ability to remember the correct information right before making a navigational choice. We report that rats

**Table 1.** Overview animals NMTS.

| Animal | N sessions | distruption trials | stim.control trials | no stim. trials |
|--------|-----------|--------------------|---------------------|-----------------|
| LD06   | 8         | 34                 | 32                  | 24              |
| LD07   | 7         | 27                 | 30                  | 29              |
| LD08   | 7         | 31                 | 31                  | 32              |
| LD12   | 6         | 40                 | 39                  | 43              |
| LD13   | 4         | 15                 | 21                  | 24              |
| Total  | 32        | 147                | 153                 | 152             |

had an equally high performance with or without awake SWR disruption, which suggests that the timeframe of impact on behavior of awake SWR-associated replay is not immediate or intermediate but most likely long-term (>1 hr).

## Results

### Awake sharp-wave ripple disruption does not affect performance on a single trial non-match-to-sample task

In the first set of experiments, five rats performed a hippocampus-dependent spatial non-match-to-sample task (NMTS, *Table 1*) in an eight-arm radial maze (*Packard et al., 1989*; *Sasaki et al., 2018*). Each trial in the task is independent and consists of an instruction (encoding) and test (retrieval) phase, separated by several seconds. The NMTS task tests if rats can memorize visited locations in the instruction phase for the duration of the trial (~1 min) and use this memory to navigate to the remaining unvisited locations in the test phase (*Figure 1a and b*). Prior to SWR manipulations, all rats were familiarized with the task until they reached a learning criterion of more than 50% of trials without errors in one session (15 trials per session) for 3 days in a row (*Figure 1c*). Note that each trial featured a randomly chosen set of locations for the instruction phase that was different from the previous trials.

Following completion of the training, SWRs were detected in the extracellular field potentials recorded from hippocampal area CA1 (*Buzsáki, 2015*; *Ciliberti and Kloosterman, 2017*). In line with previous reports (*Singer and Frank, 2009*), most ripples occurred during reward consumption and immobility periods in the instruction and test phases (ripple rate, offline detected, instruction: 0.66 Hz [0.55,0.79], test: 0.58 Hz [0.47,0.71]).

In one third of the trials in every session, ripple detection was used to trigger closed-loop electrical stimulation of the ventral hippocampal commissure to transiently disrupt the ongoing SWR (same method as *Michon et al., 2019*; *Figure 1d*). Note that rats at this point have learned the general task rules, but still need to remember a new random set of trial-specific locations. The task performance of the rats in the disruption trials was compared to the performance in the remaining control trials with either delayed stimulation upon ripple detection (random delay between 150 and 250ms) or without stimulation. Ripple detection and disruption happened during all phases of the trial (instruction and test). The trial performance (i.e. whether or not a trial is completed without a single error) and number of correct visits in the test phase in the disruption trials were not significantly different from those in the control trials (performance; disruption: 0.68 [0.57,0.77], stim.control: 0.73 [0.63,0.81], no stim. 0.72 [0.62,0.80], Kruskal-Wallis test: H=0.93, p=0.63, correct visits test phase; disruption: 3.52 [3.30,3.67], stim.control: 3.66 [3.49,3.77], no stim.: 3.49 [3.24,3.66], H=1.17, p=0.56, *Figure 1e and f*).

For rats to correctly perform the task and identify which arms have not yet been visited, one strategy is to memorize all four visits from the instruction phase. This memory may be subject to temporal decay such that the last visited arm in the instruction phase is remembered best. Indeed, we observed a temporal order effect such that errors in the test phase are biased to the arms that were instructed furthest back in time (average order number instruction arm; disruption: 1.70 [1.45,2.02], stim.control: 1.93 [1.60,2.29], no stim.: 1.77 [1.47,2.11], *Figure 1—figure supplement 1c*). However, we observed no difference between trials with or without ripple disruption (Kruskal-Wallis test: H=1.84, p=0.4,

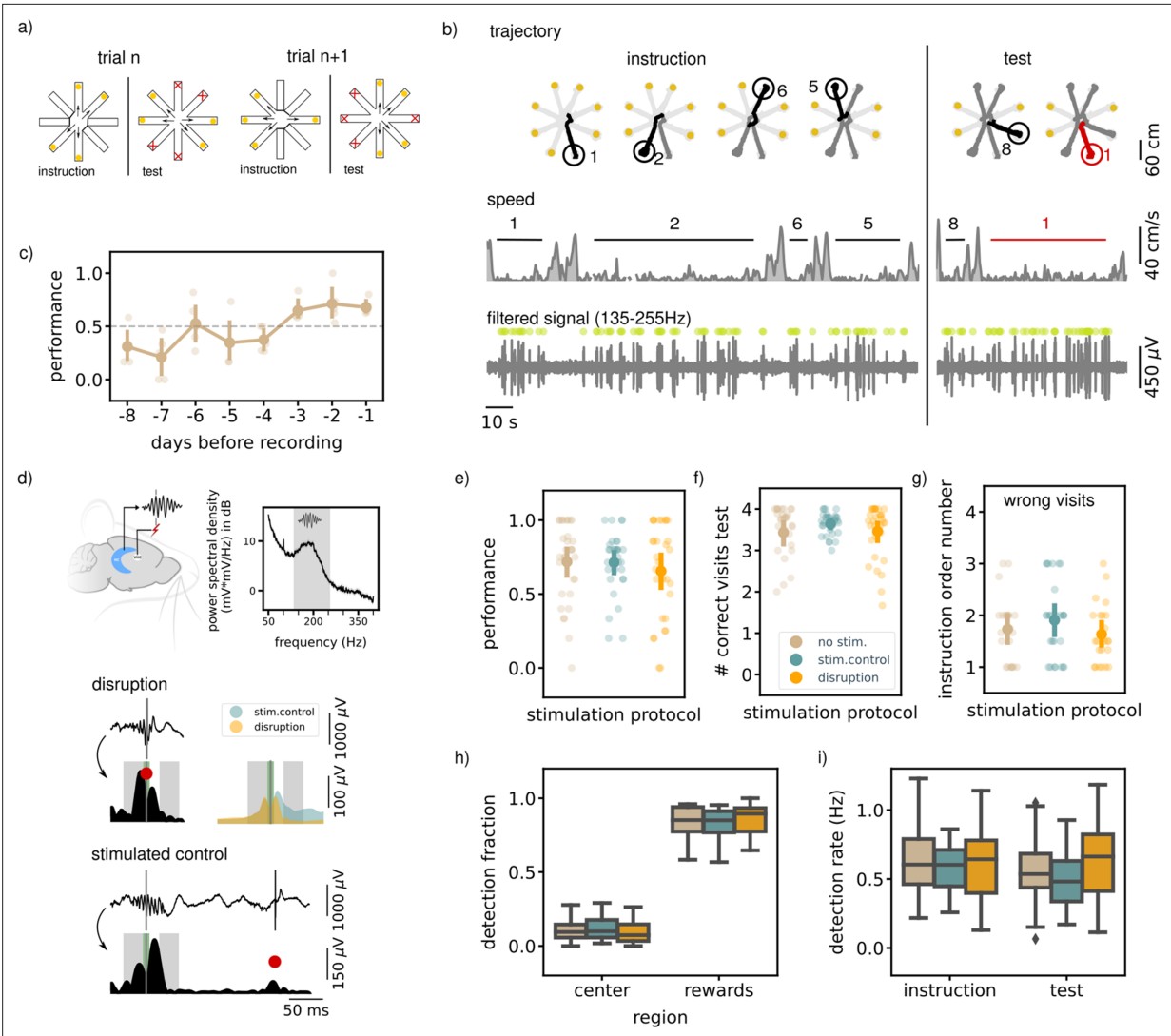

**Figure 1.** Executing a non-match-to-sample (NMTS) task is not awake SWR dependent. (**a**) A schematic illustration of the setup used for the behavior. In a NMTS task all eight arms are rewarded but only four can be collected in the instruction phase and the rest needs to be collected in the test phase without an error. (**b**) Trajectory, speed and filtered LFP for an example trial. Red trajectory indicates a wrong visit, yellow dot a to be collected reward and the green dot an online ripple detection. (**c**) The learning curve over pretraining days for all animals (N=5) showing the average trial performance in a session. The dashed line indicates the learning criteria (small dots; results for individual animals, large dots; mean and 99% CI). (**d**) (top) Illustration showing the recording (HC) and stimulation (VHC) sites and a power spectral density of an example control session considering only the immobility periods (speed <5 cm/s). The grey shaded area indicates the chosen frequency range to filter the LFP before ripple detection. (bottom) Example LFP and envelope traces of a disrupted and delayed stimulated (bottom) ripple. The small grey line indicates the time of detection, the red dot the point of stimulation. The green shaded area covers the window in which the stimulation artifact was removed and the grey shaded areas indicate the time windows used to compare the ripple envelope before and after ripple detection. The top right shows the average envelope of all disrupted and delay stimulated ripples in one example session. (**e**) The average trial performance in a session per stimulation protocol (no stim: n=169, stim. control: n=155 and disruption: n=148). (**f**) Same as (**e**) but showing the number of visits in the test phase. (**g**) The instruction phase order number of the wrongly visited arm per stimulation protocol. Only wrong trials are considered. (no stim: n=47, stim. control: n=42and disruption: n=47). For visualization purpose each small dot in **e**, **f** and **g** represents a session average. The large dots represent the mean and 99%CI per condition. (**h**) The fraction of ripple detections during the trials on the rewards and central platforms. The small remaining fraction of detections happened on the arms. (**i**) The ripple detection rate during the instruction and test phase of the trials, considering only periods of immobility (speed <5 cm/s). In **h** and **i** the box shows the quartiles of the dataset while the whiskers extend to show the rest of the distribution. Outliers (defined using the inter-quartile range) are shown using a diamond marker. See also *Figure 1—figure supplement 1*.

The online version of this article includes the following figure supplement(s) for figure 1:

**Figure supplement 1.** Additional behavioral analyses NMTS task show no effect of ripple disruption.

*Figure 1g*), which taken together with the previous quantifications strongly suggests that performing the NMTS task is not impaired by SWR disruption.

It is possible that the effect of ripple disruption is not reflected in task performance measures, but might be observable in other behavioral parameters. For example, rats could linger on the central platform or arms for longer as an indication of their uncertainty about which arms to visit. For this, we first looked at the running speed at the central platform and arms across all three stimulation conditions. Overall, speed was lower on the central platform than on the arms. When running speed was analyzed separately for arms and center platform, no significant difference was found between the three stimulation protocols (*Figure 1—figure supplement 1a*). Likewise, the time per visit in the test phase was not impacted by ripple disruption (*Figure 1—figure supplement 1b*).

Due to a difference in experimental setup (see Methods), the inter-phase time for three animals was longer and more varied compared to the inter-phase times in the remaining two animals (*Figure 1— figure supplement 1e*). To test if trials with longer intervals may be more susceptible to ripple disruption, we compared the task performance between both groups of animals. We observed a tendency for rats to perform better with short inter-phase intervals, but there was no difference between stimulation conditions for either short or long intervals (*Figure 1—figure supplement 1f*).

To look for possible compensatory effects of ripple disruption on network activity, we analyzed the location of ripple detections. In line with the literature, the largest fraction of ripples were detected when the rats were on the reward platform (rewards; disruption: 0.85 [0.79,0.90], stim.control: 0.82 [0.77,0.87], no stim.: 0.84 [0.78,0.88], center; disruption: 0.10 [0.06,0.14], stim.control: 0.12 [0.09,0.16], no stim.: 0.10 [0.07,0.14], arms; disruption: 0.05 [0.03,0.08], stim.control: 0.06 [0.04,0.09], no stim.: 0.06 [0.04,0.08], *Figure 1h*). The bias for ripple detection on the reward platform was not significantly different between the different stimulation protocols (bias computed as the difference in ripple detections on the reward and central platforms divided by the total number of detections; Kruskal-Wallis test: H=1.53, p=0.46), showing that location of the ripple detections did not change across stimulation conditions. We also did not observe a difference in bias when looking at the instruction or test phase separately (*Figure 1—figure supplement 1d*).

The spatial distribution of ripple occurrence was not influenced by ripple disruption overall (Kruskal-Wallis test: H=1.53, $P$=0.46), nor in either instruction or test phase (*Figure 1—figure supplement 1d*). The online ripple detection rate, calculated using only periods of immobility (run speed <5 cm/s) in the trials was also not significantly different between the stimulation protocols in either the instruction and test phase (instruction: disruption: 0.61 Hz [0.49,0.72], stim.control: 0.58 Hz [0.50,0.65], no stim.: 0.64 Hz [0.54,0.76], Kruskal-Wallis test: H=0.73, p=0.7, test: disruption: 0.63 Hz [0.50,0.75], stim.control: 0.49 Hz [0.41,0.58], no stim.: 0.55 Hz [0.44,0.67], Kruskal-Wallis test: H=4.29, p=0.12, *Figure 1i*).

## Awake sharp-wave ripple disruption does not affect performance on a multiple trial match-to-sample task

In a second set of experiments, five rats performed a hippocampus-dependent spatial match-to-sample task (MTS, *Table 2*) in an eight-arm radial maze (*Okaichi and Oshima, 1990*; *Figure 2a*). In this task, rats were required to memorize and return to a number of fixed rewarded locations across repeated trials. In daily training sessions, four randomly selected arms were baited and rats had to learn over the course of 25 trials to only visit the four baited arms in each trial. As learning progressed

**Table 2.** Overview animals MTS.

| Animals | Disruption sessions | Stim.control sessions | No stim sessions |
|---------|--------------------|-----------------------|------------------|
| LD10 | 4 | 3 | 1 |
| LD11 | 3 | 2 | 3 |
| LD14 | 9 | 10 | 6 |
| LD21 | 8 | 8 | 4 |
| LD22 | 7 | 8 | 3 |
| Total | 31 | 31 | 17 |

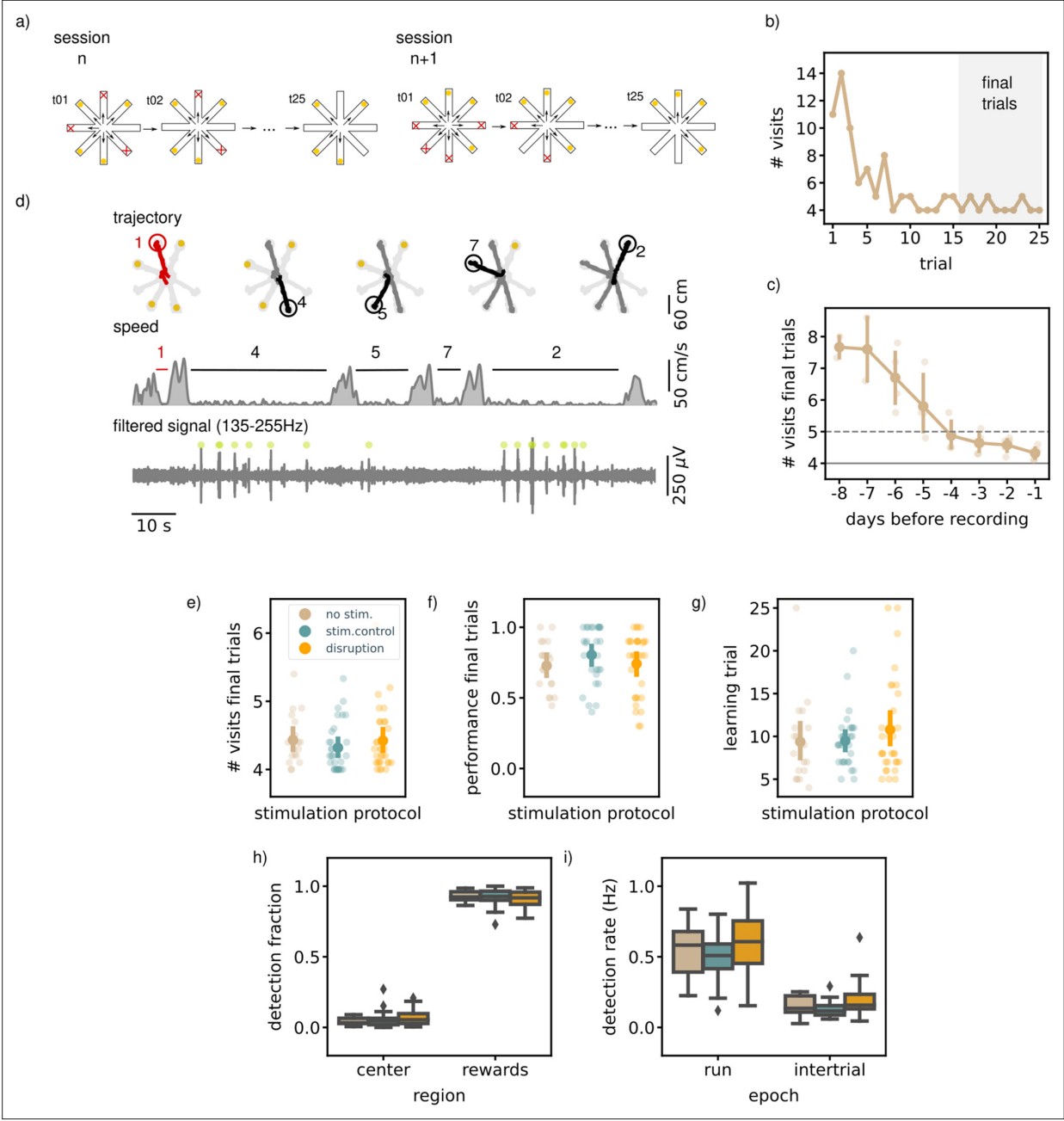

**Figure 2.** Executing a match-to-sample task (MTS) is not hippocampal ripple dependent. (**a**) A schematic illustration of the setup used for the behavior. In an MTS task only four of the eight arms are rewarded. (**b**) Learning curve of one example session showing the total number of visits in each trial. (**c**) The learning curve over pretraining days for all animals (N=5) showing the average number of visits in the final trials per session (small dots; results for individual animals, large dots; mean and 99% CI over animals). The dashed line indicates the learning criteria, the solid line the perfect performance. (**d**) Trajectory, speed and filtered LFP for an example trial. Red trajectory indicates a wrong visit, yellow dot a to-be-collected reward and the green dot an online ripple detection. (**e**) The average number of visits in the final trials per stimulation protocol. (**f**) The average performance (binary quantification) in the final visits per stimulation protocol. (**g**) The learning trial per stimulation protocol. In **e**, (**f**, and **g**) small dots represent results for individual sessions (disruption: n=32, stim.control: n=31, no stim.: n=19), large dots reflect the mean and 99%CI per stimulation protocol. (**h**) The fraction of ripple detections during the trials on the rewards and central platforms. The small remaining fraction of detections happened on the arms. (**i**) The ripple detection rate during trials considering only periods of immobility (speed <5 cm/s) and during inter-trial epochs. In **h** and **i** the box shows the quartiles of the dataset while the whiskers extend to show the rest of the distribution. Outliers (defined using the inter-quartile range) are shown using a diamond marker. See also *Figure 2—figure supplement 1*.

The online version of this article includes the following figure supplement(s) for figure 2:

**Figure supplement 1.** Additional behavioral analyses MTS task show no effect of ripple disruption.

within each session, rats made fewer excursions to non-rewarded arms and the number of arm visits per trial decreased (*Figure 2b and d*). Rats were trained until the average number of visits per trial for the final 10 trials met the learning criterium (<5 visits) consistently for three sessions in a row (*Figure 2c*).

Once the rats reached the learning criterion and were familiar with the general task to memorize a new set of locations every session, awake ripples were disrupted in one group of sessions (N=32). The task performance of the rats in these disruption sessions was compared to the performance in the remaining control sessions with either delayed stimulation upon ripple detection (N=31) or without stimulation (N=19). Detection and disruption of ripples was performed during all trials and intervening inter-trial intervals. As expected, during the trials most ripples occurred during periods of immobility (ripple rate, offline detected: mean [99% CI], 0.52 Hz [0.43,0.61]) and reward consumption (*Figure 2d*). The ripple rate in the inter-trial periods was lower than during the trials (mean [99% CI], 0.14 Hz [0.10,0.18]).

The ability of rats to learn the location of the four baited arms was not affected by SWR disruption. In the final 10 trials at the end of each session, neither the average number of arm visits to collect rewards (mean [99% CI]; disruption: 4.42 [4.27,4.71], stim. control: 4.32 [4.18,4.52], no stim.: 4.43 [4.27,4.71], Kruskal-Wallis test: H=2.43, p=0.3, *Figure 2e*) nor the fraction of trials without any error (disruption: 0.74 [0.64,0.83], stim. control: 0.80 [0.71,0.88], no stim.: 0.73 [0.63,0.83], Kruskal-Wallis test: H=2.56, p=0.28, *Figure 2f*) differed between disruption and control conditions. To test if the rate of learning over the course of a session was influenced by ripple disruption, we defined a learning trial as the first trial following three correct trials in a row. Rats learned the location of the rewarded arms equally fast in all conditions (learning trial, disruption: 10.78 [8.66,13.94], stim. control: 9.48 [8.26,11.33], no stim.: 9.37 [7.32,13.34], Kruskal-Wallis test: H=0.70, p=0.71, *Figure 2g*).

We quantified other behavioral parameters to investigate a possible impact of ripple disruption. Similar to the results in the NMTS task, rats ran faster on the arms than on the central platform suggesting that rats take time on the central platform to select the next arm to visit. If SWRs are important for identifying the baited arms, rats might be more uncertain about their choice in disruption sessions which could be reflected in increased lingering at the central platform. We observed no difference in running speed between between the three stimulation protocols (see *Figure 2—figure supplement 1a*). To assess in more detail if rats showed hesitation to perform the task, we divided each session into three stages (early [1-8], middle [9-17], and late [18-25] trials) and looked at the average time spent per arm visit. For all three stages, the average time per visit was not significantly different for the disruption condition as compared to the control conditions (*Figure 2—figure supplement 1b*).

Next, we considered that rats could employ a different strategy in the disruption sessions, which could serve as a compensatory mechanism for the missing SWRs. One well-known strategy for solving a match-to-sample task in an eight-arm radial maze is searching for rewarded arms in clockwise or counterclockwise order (*Okaichi and Oshima, 1990*). Rats did not engage in circular choice behavior more often than would be expected by chance for any of the stimulation conditions, and a direct comparison between ripple disruption and control conditions did not reveal a change in circular choice behavior (*Figure 2—figure supplement 1c*). We noticed that once the rats knew which four arms delivered reward, they tended to visit these arms each time in the same order. To investigate if SWR disruption influenced this stereotypical behavior, we computed a stereotypy index for every session by computing the average pairwise similarity between visit sequences of the final trials. The similarity measure was based on the Levenshtein distance (*Levenshtein, 1996*) that equals 0 if the visit sequences are the same, and adds 1 for every deletion, insertion or substitution necessary to map one visit sequence onto another. As expected, the stereotypy index was on average higher than expected by chance (disruption: 0.62 [0.55,0.70], stim.control: 0.65 [0.56,0.75], no stim.: 0.59 [0.51,0.68], *Figure 2—figure supplement 1c*) but was not significantly different between disruption and control sessions (Kruskal-Wallis test: H=0.64, p=0.72).

In all of the above analyses, we assumed that each configuration of four arms is equally difficult to learn. To confirm this assumption, we split the configurations in three different categories; (IIa) two sets of neighboring arm pairs, (IIb) one pair of neighboring arms and two arms without direct neighbors and (III) three neighboring arms and one arm without direct neighbors. For this analysis, we left out the sessions in which all four arms have no direct neighbor due to a low sample size (n=2 for each

condition). For all three arm configurations, ripple disruption had no influence on any performance quantification, see *Figure 2—figure supplement 1d*.

Similar to the NMTS task, the largest fraction of ripples was detected when the rats were on the reward platforms (disruption: 0.91 [0.88,0.93], stim.control: 0.92 [0.89,0.94], no stim.: 0.93 [0.91,0.95]) followed by the center (disruption: 0.07 [0.05,0.09], stim.control: 0.05 [0.04,0.09], no stim.: 0.04 [0.03,0.06]) and arms (disruption: 0.02 [0.02,0.04], stim.control: 0.02 [0.01,0.03], no stim.: 0.03 [0.02,0.05], *Figure 2h*). The spatial distribution of ripple occurrence was not influenced by ripple disruption (Kruskal-Wallis test: H=1.80, p=0.41). The online detection rate in the trial or inter-trial periods, calculated using only periods of immobility (run speed <5 cm/s), was not different in disruption sessions versus control sessions (trial; disruption: 0.62 Hz [0.52,0.78], stim.control: 0.50 Hz [0.42,0.56], no stim.: 0.56 Hz [0.43,0.66], H=4.71, p=0.095, inter-trial: disruption: 0.19 Hz [0.15,0.27], stim.control: 0.13 Hz [0.11,0.16], no stim.: 0.15 Hz [0.11,0.19], H=5.90, p=0.052, *Figure 2i*).

## Awake sharp-wave ripple disruption does not affect performance on a continuous sequence task

The results from the first two experiments suggest that SWRs do not support the memorization of a set of locations within a short timeframe (single trial or session). Previous ripple manipulation studies used a spatial alternation task in which rats were required to go from a center arm to two outer arms in an alternating fashion, that is left, center, right, center, left,… To learn this rule, it is crucial to discover and remember the correct temporal order of arm visits, something that was not required in either the MTS or NMTS task.

To test if SWRs are required for memorizing the temporal order of two or more recent events, we designed a spatial sequence memory paradigm (*Figure 3a*). In this task, rats need to learn each day the order in which to visit a set of four pseudo-randomly selected arms by remembering which arm-arm transitions are rewarded. After extensive pre-training on the linear track and a three-arm radial maze, five rats were pre-trained on the sequence memory paradigm (*Table 3*) for several days until the final average sequence performance (average sequence performance in the last 100 arm visits) was above 50% for at least two sessions in a row. The sequence performance is calculated by multiplying the outcome of all individual visits (1=correct transition, 0=incorrect transition) in a moving window of five visits. A correct sequence performance (i.e. five correct visits in a row), will yield a sequence performance of one (*Figure 3b and c*).

To test the involvement of awake SWRs in memorizing and learning a spatial sequence, we disrupted ripples in one third of the sessions (N=23), throughout the two run epochs and the rest epoch. The other sessions were used for stimulated (N=29) and non-stimulated (N=31) controls, similarly as defined for the MTS and NMTS task. Ripples were mostly detected on the reward platform when the animal is immobile (ripple rate, offline detected, 0.47 Hz [0.41,0.53], *Figure 3c*). The ripple rate during the interleaving rest epoch is comparably high (ripple rate, offline detected, 0.58 Hz [0.53,0.64]).

Quantification of the average visit and sequence performance in the final visits indicated no significant difference between the stimulation protocols (visit performance; disruption: 0.85 [0.71,0.94], stim.control: 0.87 [0.74,0.93], no stim.: 0.85 [0.73,0.91], Kruskal-Wallis test: H=2.61, p=0.27, sequence performance; disruption: 0.67 [0.44,0.83], stim.control: 0.71 [0.53,0.83], no stim.: 0.63 [0.47,0.75], Kruskal-Wallis test: H=2.55, p=0.28, *Figure 3e and f*). To test if rats learn at a different rate, we computed for every session the smoothed learning curve of both the visit and sequence performance and defined the visit where the curve is significantly above 0.5 as the learning visit. In every stimulation condition, there was a similar percentage of sessions in which the rat does not reach the learning criteria, which we left out for this quantification (visit performance; disruption: n=5, stim. control: n=3, no stim: n=4, sequence performance; disruption: n=8, stim. control: n=10, no stim: n=11). The learning visit for both the visit and sequence performance was not significantly different between sessions with a different stimulation protocol (learning visit performance; disruption: 117.72 [74.64,173.65], stim.control: 112.69 [76.92,154.99], no stim.: 154.22 [109.28,201.30], Kruskal-Wallis test: H=2.66, p=0.26, learning visit sequence performance; disruption: 238.80 [189.36,279.90], stim. control: 209.16 [156.94,263.22], no stim.: 273.15 [202.33,315.75], Kruskal-Wallis test: H=6.01, p=0.05, *Figure 3g and h*).

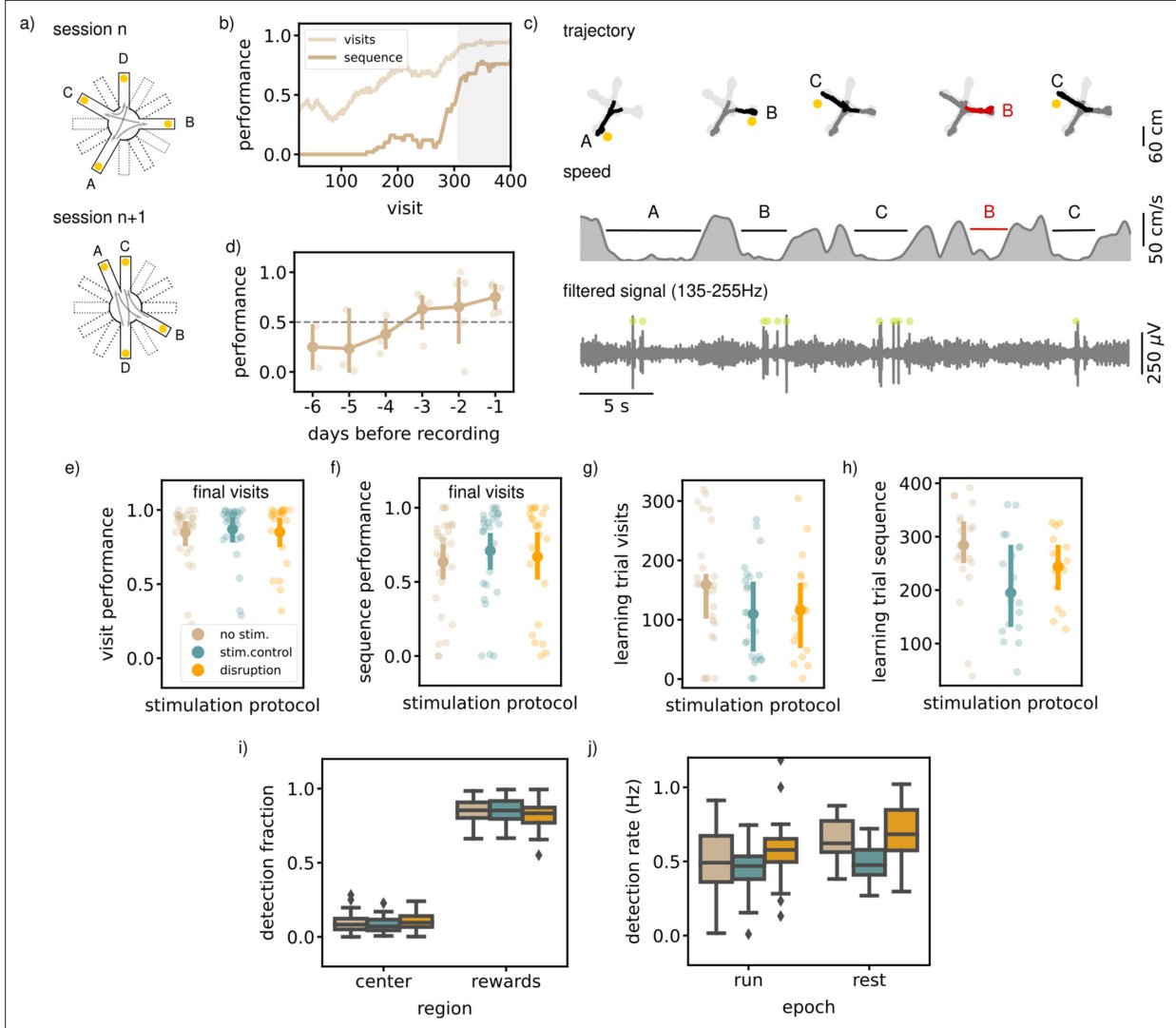

**Figure 3.** Executing a sequential (SEQ) spatial memory task is not hippocampal ripple dependent. (**a**) A schematic illustration of the setup used for the behavior. In a SEQ task only four of twelve possible arms are present. Reward is delivered when arms are visited in the correct order. (**b**) Learning curves of one example session showing the visit performance and sequence performance over visits, both visualized using a moving average with N=50. (**c**) Trajectory, speed and filtered LFP for five visits in an example session. Red trajectory indicates a wrong visit, yellow dot a to be collected reward and the green dot an online ripple detection. (**d**) The learning curve over pretraining days for all animals (N=5) showing the average sequence performance in the final 100 visits in a session. The dashed line indicates the learning criteria (small dots; result per animal, large dots: mean and 99% CI). (**e**) The average visit performance over the final visits of a session per stimulation protocol. One small dot represents one session (no stim: n=31, stim. control: n=29 and disruption: n=23) and the large dot the mean and 99%CI. (**f**) Same as (**e**) but showing the average sequence performance over the final visits of a session. (**g**) The visit learning trial of sessions per stimulation protocol defined using the smoothed learning curve of the visit performance (see methods). Only session where learning criteria was met within 400 visits are included. (**h**) Same as (**g**) but showing the sequence learning trial computed using the smoothed learning curve of the sequence performance. (**i**) The fraction of ripple detections during the trials on the rewards and central platforms. The small remaining fraction of detections happened on the arms. (**j**) The ripple detection rate during run on the maze, considering only periods of immobility (speed <5 cm/s), and during rest epochs in the sleep box. In **i** and **j** the box shows the quartiles of the dataset while the whiskers extend to show the rest of the distribution. Outliers (defined using the inter-quartile range) are shown using a diamond marker.See also *Figure 3— figure supplement 1*.

The online version of this article includes the following figure supplement(s) for figure 3:

**Figure supplement 1.** Additional behavioral analyses SEQ task show no effect of ripple disruption.

**Table 3.** Overview animals SEQ.

| Animals | Disruption sessions | Stim. Control sessions | No stim. sessions |
|---|---|---|---|
| LD53 | 7 | 7 | 7 |
| LD56 | 8 | 7 | 6 |
| LD57 | 5 | 6 | 6 |
| LD58 | 5 | 5 | 6 |
| LD60 | 6 | 4 | 4 |
| Total | 31 | 29 | 29 |

Quantification of general behavioral parameters did not reveal any change due to ripple disruption. Like in the previous experiments, we quantified the speed on the central platform and arms and looked at the time per visit as a measure for how certain rats were about their choice. The speed on the central platform and arms showed no difference in different stimulation conditions (*Figure 3—figure supplement 1a*). Rats did not take more or less time per visit in disruption versus control trials (*Figure 3—figure supplement 1b*).

When we looked at the smoothed learning curves for the individual arm-arm transitions, we observed that not all transitions were learned equally fast (*Figure 3—figure supplement 1c*). This is an indication that rats are probably not learning individual transitions, but are aware that there is a repeating pattern to be found. Namely, the possibilities for the new transitions are lower when using the information of learned transitions in the correct temporal order. By looking at the learning visits for the individual transition learning curves, we can investigate if the strategy employed by the rats changes due to ripple disruption. For all three stimulation protocols, learning visits of individual transitions vary and the distributions were significantly different between stimulation protocol (*Figure 3—figure supplement 1d*). Post-hoc Mann-Whitney tests, however, revealed only a significant difference between the two control conditions (*Figure 3—figure supplement 1d*). Next, we quantified the difference between the learning visits of the slowest and fastest learned transition in sessions where the rat had learned at least two transitions, as a measure of how much use had been made of the repeating pattern. This quantification of the learning trial difference in each session showed no significant difference due to ripple disruption (*Figure 3—figure supplement 1e*). Together these quantifications suggest that SWR disruption has no overall effect on the behavior in the sequence task.

In line with our expectations, ripples were detected mostly at reward sites (rewards; disruption: 0.82 [0.78,0.85], stim.control: 0.85 [0.82,0.87], no stim.: 0.85 [0.82,0.87], arms; disruption: 0.08 [0.06,0.10], stim. control: 0.07 [0.06,0.09], no stim.: 0.06 [0.05,0.07], center; disruption: 0.10 [0.09,0.12], stim. control: 0.08 [0.07,0.10], no stim.: 0.09 [0.08,0.11], *Figure 3i*), and this bias was not significantly different between stimulation protocols (Kruskal-Wallis test: H=0.33, p=0.85). The overall ripple detection rate calculated over the immobility periods in both run epochs and in the rest epochs did appear to be significantly different between sessions with a different stimulation protocol (RUN; disruption: 0.58 [0.52,0.67], stim.control: 0.46 [0.41,0.50]), no stim.: 0.49 [0.41,0.57], Kruskal-Wallis test: H=12.94, p=0.0016, (REST; disruption: 0.69 [0.58,0.78], stim. control: 0.49 [0.43,0.55]), no stim.: 0.64 [0.58,0.70], (Kruskal-Wallis test: H=21.02, p=$2.7 \times 10^{-5}$). Post-hoc Mann-Whitney tests for the run and rest epochs revealed a significant difference between the disruption and stimulated control sessions (run and rest) or between the stimulated control and non-stimulated control session (rest) and not between the other pairs (RUN: disruption - stim.control: U=1944.00, p=$6.6 \times 10^{-5}$, disruption - no stim.: U=1728.00, p=0.061, stim.control - no stim.: U=2011.00, p=0.26 REST: disruption - stim.control: U=520.00, p=0.00019, disruption - no stim.: U=435.00, p=0.17, stim.control - no stim.: U=695.00, p=$7.7 \times 10^{-5}$).

## Awake sharp-wave ripples are accurately detected and disrupted

All experiments were conducted using the same implementation for online ripple detection and disruption as in our previous work (*Michon et al., 2019*), where it was shown that ripple-triggered VHC stimulation following learning reduced spatial memory for highly rewarded places. Here, we use the same

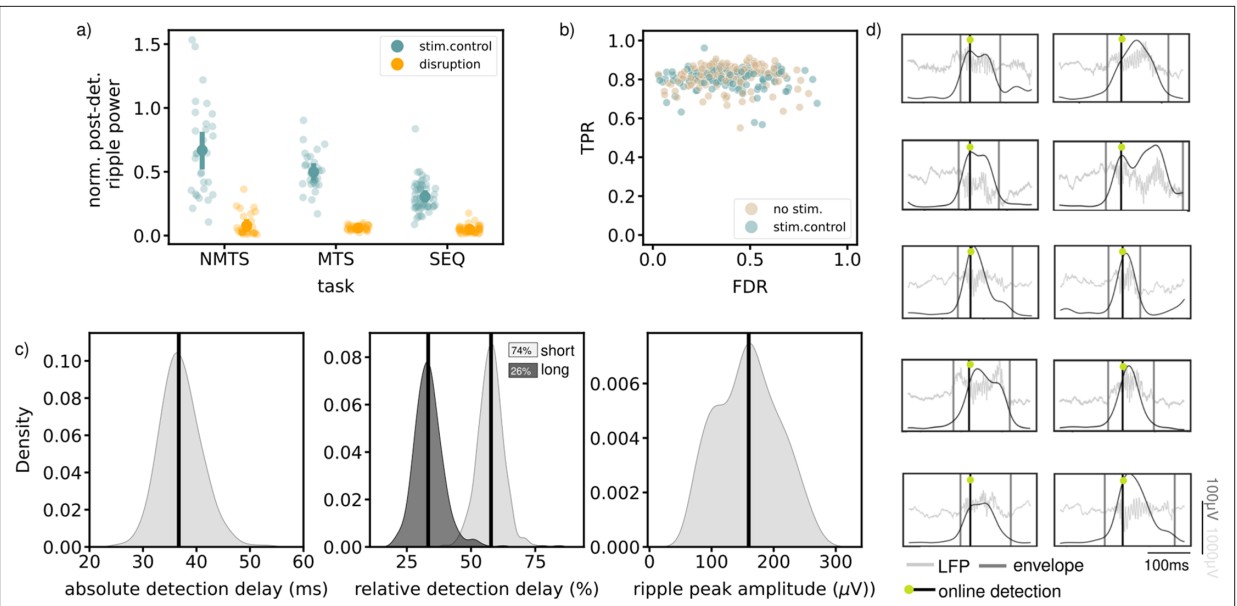

**Figure 4.** Available quantifications indicate accurate ripple detection and disruption. (**a**) Normalized post-detection ripple power for all tree tasks in run epochs (NMTS: disruption: n_trials=148, stim.control: n_trials=155 , MTS: disruption: n_sessions=31, stim.control: n_sessions=31, SEQ: disruption: n_sessions=23, stim.control: n_sessions=29). A small dot represents the average over one session, the large dots represent the mean and 99%CI. (**b**) True positive rate versus False discovery rate for all tasks. One dot represents the average over one session. (**c**) The absolute detection delays (left), relative detection delays for both the short (74% of all ripples) and long ripples (26% of all ripples) (middle) and ripple peak amplitude for all epochs of all tasks, considering the no stim. and stim.control epochs. (**d**) Example online detected ripples that were also detected offline (true positive) from five different animals. Grey vertical lines indicate the start and end of the ripple. See also *Figure 4—figure supplements 1 and 2*.

The online version of this article includes the following figure supplement(s) for figure 4:

**Figure supplement 1.** Hippocampal ripples are disrupted bilaterally.

**Figure supplement 2.** Quantification of ripple detection and disruption in rest epochs.

post-hoc quality metrics for both the disruption and detection to show that we indeed performed the manipulations in a way that is comparable to those that impacted behavior in a different scenario.

Before each experimental session, the threshold for ripple detection and the stimulation current were manually adjusted by the experimenter to ensure accurate and complete detection and disruption. The disruption quality was quantified for all sessions in stimulated control and disruption condition by comparing the ripple amplitude in a 20ms window before and after the time of detection. The normalized post-detection ripple amplitude was significantly lower in the sessions with ripple disruption in all three tasks (Mann-Whitney test, NMTS: U=912.00, p=$5.9 \times 10^{-11}$, MTS: U=961.00, p=$7 \times 10^{-12}$, SEQ: U=2615.00, p=$2.6 \times 10^{-18}$), showing that detected ripples were successfully disrupted (*Figure 4a*).

Since the stimulation electrode is placed in the ventral hippocampal commissure, a large white matter structure constructed from axons connecting the left and right hippocampi, ripples are disrupted in both hemispheres. Indeed, recordings from three rats with recording tetrodes placed in both the left and right hemisphere showed a clear disruption of ripples in both hemispheres using the

**Table 4.** Overview animals bilateral recordings.

| Animals | Disruption sessions | Stim. Control sessions |
|---|---|---|
| ST3 | 3 | 3 |
| ST4 | 4 | 4 |
| ST5 | 4 | 4 |
| Total | 11 | 11 |

same unilateral stimulation method as in the three behavioral experiments (*Table 4*; *Figure 4—figure supplement 1*).

The quality of the ripple detection was quantified based on data from the stimulated and non-stimulated control sessions. In each session, ripples were detected offline (see Methods) and compared to the online detected ripples by quantifying a true positive rate (TPR) and false discovery rate (FDR) (*Figure 4b*). The high TPR for almost all sessions suggest that a large fraction of all offline-defined ripples were detected online. We observed a wider distribution for the FDR in comparison to the distribution for the TPR, likely due to a variable signal-to-noise ratio across experiments. Ripple detection and disruption were quantified separately for the inter-trial times in the MTS task and the rest epochs in the SEQ task (*Figure 4—figure supplement 2*). In both cases, ripple power was significantly reduced following stimulation (*Figure 4—figure supplement 2a*). The TPR-FDR distribution for the inter-trial periods in the REF task was shifted toward lower TPR and higher FDR, indicating a worse detection (lower TPR and higher FDR, *Figure 4—figure supplement 2b*). In the inter-trial times, the animals were locked in the center of the maze and they frequently engaged with the closed doors, which resulted in movement artifacts in the LFP signal that were falsely identified as ripples. Quantifications for the sleep box epochs in the SEQ task were similar to those on the maze, that is high TPR and variable FDR (*Figure 4—figure supplement 2c*). As online detection of ripples can only occur after their initiation, there will always be a detection delay (*Figure 4c*). The absolute detection delay over all epochs in each task, and considering only the no stim. and stim.control epochs, is median [iqr], 36.69ms [34.68,39.19], in line with what is expected when using the Falcon software platform (*Ciliberti and Kloosterman, 2017*). When we compare this to the total length of the ripple, we obtain a relative delay (median [iqr]) of 33.32% [29.97,36.47] for long ripples (duration >100ms) and 57.79% [54.61,60.49] for the remaining ripples. The average peak ripple amplitude, quantified by filtering in the ripple frequency band (140–225 Hz) and taking the peak value of the envelope in the ripple interval, is 159.76 µV [117.61,195.45].

## Discussion

Altogether, our results show with direct causal evidence that awake SWRs do not support the memorization of a set of locations within a short timeframe (single trial or session) nor do they support the memorization of the temporal order of two or more recent events. Our results also revealed that ripple disruption had no influence on other behavioral parameters or on the frequency and location of ripple occurrence. Since SWRs often coincide with replay events, our results suggest that awake hippocampal replay is not pertinent for the immediate future behavior, nor for the temporary memorization of relevant events at an intermediate time scale (seconds to minutes), as was previously hypothesized based on correlative evidence (*Singer et al., 2013*; *Ólafsdóttir et al., 2017*; *Shin et al., 2019*; *Gillespie et al., 2021*; *Ambrose et al., 2016*; *Diba and Buzsáki, 2007*; *Foster and Wilson, 2006*; *Karlsson and Frank, 2009*). We hypothesize that the timeframe of impact for awake SWRs is long-term and cumulative over time, potentially supporting long-term consolidation processes.

For all three experiments in this study, we purposely selected repeated acquisition spatial memory tasks that were designed to separate the acquisition of general context and task information from the memorization of new sets of locations in each trial or daily session. These task designs allowed us to specifically test the contribution of hippocampal SWRs to within-trial and within-session memory demands. Both the NMTS and MTS tasks have been shown to depend on an intact hippocampus (*Sasaki et al., 2018*; *Okaichi and Oshima, 1990*). The SEQ task is new but has a high similarity to the three arm alternation task that has been shown to be hippocampal dependent (*Racine and Kimble, 1965*), and a number of studies indicate that the hippocampus is important for remembering the temporal order of events (*Gilbert and Kesner, 2002*; *Fortin et al., 2002*; *Kesner et al., 2005*; *Farovik et al., 2010*). Although it could be argued that repeated training in the same task (albeit with randomly varying reward locations) may reduce or abolish the dependence on the hippocampus, for at least two spatial memory tasks - the NMTS task used in this study (*Sasaki et al., 2018*) and place learning in the Morris water maze (*Blokland et al., 1992*) - continued hippocampal-dependence has been demonstrated. However, we cannot completely exclude that rats may have acquired an optimized and less cognitively demanding strategy that is mainly dependent brain structures outside of the hippocampus.

Our observation that SWR disruption did not affect performance in any of the tested tasks, suggest that SWRs do not support within-session or within-trial spatial memory. These results are in apparent contrast to previous SWR intervention studies showing altered learning a spatial alternation rule in a three-arm maze (*Jadhav et al., 2012*; *Fernández-Ruiz et al., 2019*). These studies are often interpreted as SWRs supporting spatial 'working' memory, that is keeping in mind a recently visited location or a next goal location, because the deficit is seen specifically for phases of the task when a choice between two navigational options needs to be made. However, there exist alternative explanations, in which SWRs and associated replay act at longer timescales, that could reconcile these different findings. First, in *Jadhav et al., 2012* and *Fernández-Ruiz et al., 2019*, animals learn the alternation rule in a novel spatial and task context across multiple days. SWRs could support the gradual acquisition of spatial and task information, either directly or indirectly through stabilization of the place code (*Roux et al., 2017*) that may facilitate subsequent sleep memory consolidation. Additionally, SWRs could part of a process that integrates information across trials and sessions to link rewarding trajectories and construct an optimal run strategy. The contribution of SWRs and associated replay would then be particularly important for when there is ambiguity in which path to follow (as is the case for the outbound component in the alternation task) and not for simpler stimulus-response associations (as is the case for the inbound component). Second, the alternation task can be solved using a striatal turn-based strategy (i.e. left-left-right-right). Indeed, lesions of the (dorsal) striatum impair the acquisition of a continuous spatial alternation task (*Moussa et al., 2011*). SWRs and replay could be instrumental in shifting from a spatial hippocampal-dependent strategy to an ego-centric striatal-dependent strategy, possibly mediated by coordinated reactivation in the hippocampal-striatal network (*Lansink et al., 2008*; *Lansink et al., 2009*). There is a third possibility, namely that SWRs *do* support spatial working memory, but not universally and only in *novel* task contexts. In familiar task contexts, spatial working memory would then be supported by (non-hippocampal) cortical mechanisms. In future experiments, trial-specific SWR manipulations in the spatial alternation task could help to test this hypothesis. In general, to find out what role awake SWRs play in long-term learning, specific causal experiments need to be designed that can isolate and investigate the different aspects of learning separately.

A number of previous studies are in line with our new hypothesis that the timeframe of impact for awake SWRs is long-term or cumulative over time. The recent study by *Gillespie et al., 2021* looked at replay activity during SWRs and employed a new short-term memory paradigm that allows the clear distinction between replay of future and past paths by having many spatially and temporally distinct navigational options. In line with our results, they did not observe a correlation between replay content and upcoming navigation choices but rather observed that the replay represented seemingly non-relevant past trajectories. Another study by *Roux et al., 2017* report a remapping of place fields when hippcampal neurons are optogenetically inhibited during awake SWRs. This remapping is observed after a 1 hr rest session when testing the memory of previously learned place-reward associations. It would be interesting to see if our approach causes a more wide-spread remapping after a long rest and potentially affects spatial memory. Lastly, there are a number of interventional studies that manipulate SWRs during sleep and show a causal link with spatial memory (*Skaggs and McNaughton, 1996*; *Girardeau et al., 2009*; *van de Ven et al., 2016*; *Michon et al., 2019*; *Gridchyn et al., 2020*). Given the high physiological similarity between awake and sleep SWRs it is conceivable that they would perform the same function and enable long-term consolidation.

Despite the lack of a behavioral task that was positively affected by ripple disruption (i.e. a 'positive control'), we believe our post-hoc analyses show that the ripples were accurately detected and disrupted. The quality of the ripple detection was assessed by two factors; the fraction of detected ripples and the detection latency. Which ripples are detected depends on the employed real-time algorithm, on the signal-to-noise ratio of the recorded signals and on the used detection threshold. The ripple detection rates in our experiments are in line with what is reported in literature and a comparison of the online detections with an offline detection algorithm yielded a high true positive rate. This suggests that a large fraction of the SWRs is online detected in our experiments. The detection latency of SWRs is a result of the real-time filtering of the LFP signal which is necessary for the identification of ripples. It is unlikely however that this would be the reason for an absence of deleterious effect on the short-term memory performance as ripple disruptions with similar latencies caused behavioral effects in previous experiments (*Michon et al., 2019*). Moreover, long ripples (>100ms) contribute most strongly to memory processes (*Fernández-Ruiz et al., 2019*) and in our experiments

close to 70% of these ripples was disrupted. The quality of the ripple disruption was assessed by looking at the ripple power drop after the electrical stimulation. The power drop is comparable to studies from other labs that used the same approach (*Michon et al., 2019*; *Girardeau et al., 2009*; *Jadhav et al., 2012*; *Fernández-Ruiz et al., 2019*). Furthermore, we also showed in a supplementary experiment that the unilateral stimulation of the VHC causes a bilateral disruption in the dorsal hippocampus and that - in line with the observed ripple power drop - the MUA activity is suppressed after a stimulation. A last element to support our confidence in the accuracy of our manipulations is that previous work from our lab employing the same method for disruption of sleep ripples did show a clear behavioral effect (*Michon et al., 2019*).

In summary, our results provide clear causal evidence that the neural activity during SWRs does not support the memorization of a set of locations within a short timeframe (single trial or session) nor does it support the memorization of the temporal order of two or more recent events. Because all three employed behavioral tasks are dependent on hippocampal activity, the question begs which hippocampal activity, if not SWRs, is required for solving these tasks. Recent literature has pointed to replay-like activity occurring outside of SWRs. These are also often referred to as 'theta sequences' and occur mostly during preparatory behaviors, such as ambulation, exploration, rearing and sniffing. Recent work has pointed to their potential role in spatial memory (*Zielinski et al., 2020*; *Wang et al., 2020*; *Schmidt et al., 2019*). We believe that this might be a good candidate neural mechanism required for the short-term rehearsal. Furthermore, one can speculate that also in primates the key neural activity required for good short-term retention of new information coincides rather with the 'theta' than with the 'non-theta' brain states - generally associated with SWRs (*Mysin and Shubina, 2023*). This hypothesis is in line with a recent report showing the importance of medial-temporal lobe theta oscillations for human short-term retention in a visual task (*Kragel et al., 2021*). Promoting the 'theta' brain state might be an interesting way to improve memory retention over a short timescale. Alternatively, more research could be done on the 'non-theta-non-ripple state' as this also remains an understudied domain both in rodents and primates (*Mysin and Shubina, 2023*).

## Methods

### Experimental model and subject details

A total of 15 male Long Evans rats (supplier Janvier Labs), food restricted to 85–90% of the free-feeding weight, were used in this study. At the start of behavioral training procedures, rats were 7–10 weeks old. All rats received an implant for neural recordings and stimulation. *Tables 1–3* provides an overview of the experimental sessions for each rat. All experiments were carried out in accordance with protocols approved by KU Leuven animal ethics committee (P119/2015 and P175/2020) and in accordance with the European Council Directive, 2010/63/EU. Animals in experiment were housed separately in individually ventilated cages (IVC) with ad libitum access to water and controlled intake of standard food pellets. Health status and body weight were checked daily by the experimenters and dedicated animal care personnel. To improve the well-being of the rats, a playpen (100x55 cm) was constructed in the course of this study that allowed rats to spend spent time in an enriched environment with their (former) cage mate. The playpen can be divided into two parts separated by a transparent wall with 'sniffle holes' to encourage social interaction between the two rats without accidental damage to their implants. The enrichment consisted of a small maze with hidden food, toys, a climbing rope and ladder that provide access to an upper story with a running wheel. Experiments for the delayed non-match-to-sample and match-to-sample tasks were started prior to completion of the playpen. All five rats used to test the memory of temporally ordered information (third experiment), however, spent at least 30 min per day in the playpen.

### Method details

#### Behavioral task

In all experiments, the maze is elevated 40 cm above the ground and located in a 4x4 m black room with distinctive visual cues on all four walls.

## Non-match-to-sample paradigm (NMTS)

We use an eight-arm radial maze with a 40 cm diameter central platform and 90 cm long arms that each terminate in a 20 cm two-reward platform. There are doors at 6 cm from the central platform on each arm, which open by moving down and are controlled wirelessly from outside the room. All animals received chocolate pellets as a reward, delivered in a small food well at the end of the reward platform. The doors were controlled wirelessly in sessions of two animals, for the sessions of the other three the doors are controlled manually. The goal of this task is for rats to remember which arms in an eight-arm radial maze they had and had not visited recently (*Sasaki et al., 2018*). In each daily session, rats perform 15 independent trials. Every trial consists of two phases: an instruction phase in which rats are forced to visit four randomly selected arms one by one, and test phase in which rats have access to all eight arms and need to visit the remaining four arms to collect reward (*Figure 1a*). At the start of each trial, all eight arms are baited with a chocolate reward at the end of the reward platform. All reward platforms contain inaccessible rewards to minimize olfactory strategies. The instruction phase starts with the rat in the center of the maze with all doors closed. Next, one after the other, four randomly chosen doors are opened and the rat is allowed to run down the arm and collect the reward. After each arm visit, the corresponding door closes so that at any moment only one arm can be visited. After the fourth visit, all doors are closed, the instruction phase ends and the rat is constrained to the central platform, in case automated doors (Peira bvba) were used, or placed on a nearby platform next to the maze, when manual doors were used. After a short delay all doors open, rats are returned to the central platform if needed, and the test phase started. After a correct visit in the test phase, the door of that arm was closed for the remainder of the trial. A trial ended after rats visited all four correct arms or after the first incorrect visit. In between trials, rats either waited for 90–120 s in the center of the maze with all doors closed (in case of automated doors) or were placed on a nearby platform (in case of manually operated doors). During the inter-trial wait period, all reward wells were refilled by the experimenter.

## Match-to-sample paradigm (MTS)

We use the same apparatus as for the NMTS task with the doors controlled wirelessly for all animals. Three animals received chocolate pellets as reward, delivered in a small food well at the end of the reward platform. The remaining two animals received a liquid reward (chocolate syrup), dispensed through a pump that was also controlled wirelessly from outside the behavior room. The goal of the task is for rats to learn in daily sessions which set of four out of eight arms lead to reward. Each session consists of 25 trials and the four rewarded arms are randomly varied between sessions, such that each session is an independent assessment of the rats' ability to learn the location of reward. Configurations that have all four arms located next to each other were not used. When a solid reward is used, a trial starts with the four chosen arms baited. The other arms contain inaccessible rewards to minimize olfactory strategies. When liquid rewards are used, a pump containing chocolate syrup is placed on every arm to minimize olfactory and visual strategies. Only the pumps on the correct arms deliver reward (~2 ml) when the rat visits that arm. At the start of a session, rats are placed on the central platform, the experimenter leaves the room, and all doors open simultaneously. Rats can now explore all arms freely. After the last rewarded arm has been visited, the rats are allowed to return to the central platform, after which all doors close and the first trial ends. In case solid rewards are used, the reward wells of the same four arms are now refilled by the experimenter. Rats remain on the central platform for 60–120 s, after which all doors are opened and the next trial starts.

## Sequence memory paradigm (SEQ)

For the sequence memory paradigm we use a 12-arm radial maze with a 37.5 cm diameter central platform and 60 cm long arms that each terminate in a 22 cm two-reward platform. There is a door on each arm, 6cm from the central platform, and all animals receive a liquid reward (condensed milk, ~2 ml) that was delivered wirelessly from outside the behavior room. In the sequence memory task, the goal is for rats to learn the correct order in which four arms need to be visited to receive reward. Each daily session is a separate experiment in which four arms of a 12-arm radial maze are randomly chosen, and the remaining arms are removed. A crossing visit order is selected (i.e. clockwise and counter clockwise orders are not used), for example if arms in positions 2,3,4,11 are chosen then valid visit orders are for example 2→4→3→11→2 and 2→3→11→4→2, but not the counterclockwise order

2→11→4→3→2. Every session consists of two run epochs lasting 25 min and an intervening rest epoch lasting 10 min. At the beginning of the session, rats are placed in the center of the maze with all doors closed. The first run epoch starts when all doors open and the rat is free to move around. Each time a correct transition is made to the next arm in the sequence the rat is rewarded. After 25 min rats are allowed to return to the center, the doors are closed and rats are taken off by the experimenter and put back in the sleep box for the rest epoch. The second run epoch is identical to the first. The rats' position is tracked by an overhead camera and liquid rewards are automatically delivered (Peira bvba, Belgium) based on the tracked behavior using custom developed software in the lab (*Catanzariti et al., 2022*; https://bitbucket.org/kloostermannerflab/fklab-controller-lab copy archived at *KloostermanNERFLab, 2024*).

## Behavioral training

All animals were pretrained to run on elevated mazes for reward, habituated to automated doors (when used) and to a separate sleep box. For each of the three behavioral tasks, the pretraining and training procedures to teach the rats the task rules are described below.

## Non-match-to-sample paradigm

In the first phase of pretraining (5–7days), rats are habituated to the eight-arm radial maze and learn to look for reward at the end of the arms. On days 1–2, rats are placed in the center of the maze and allowed to freely explore for 15–30min. Chocolate rewards are scattered throughout the maze to promote exploration. On the next 2–3days, rewards are progressively restricted to only the arms and finally the reward wells at the end of each arm. Once rats are sufficiently comfortable and motivated to run for the rewards, the second phase of the pretraining starts. This phase lasts three days and rats are required to perform at least three to six trials of the NMTS paradigm. Rats are then taken off food restriction and undergo surgery. After 1 week of post-surgery recovery, the food restriction is resumed and once rats are at 85–90% of their post-surgery weight. Finally, rats run 6–15 trials per day until the average performance is above 50% for at least three days in a row.

## Match-to-sample paradigm

The pretraining prior to surgery is the same as for the NMTS paradigm, except in the second phase where rats perform daily sessions of the MTS paradigm until they complete at least 15 trials in 30 min. Training is completed after recovery from surgery when rats are back at 85–90% of their post-surgery weight. In daily sessions (25 trials), rats perform the MTS paradigm until the average number of visits on the last 10 trials is less than 5 for 3 days in a row. In total rats take on average 14 days to reach the learning criterium.

## Sequence memory paradigm

Prior to surgery, rats are trained to run back and forth on a linear track (130 cm) for liquid reward. Animals perform two 15 min sessions per day, with a 15–20 min intervening rest in the sleep box. Training continues until rats collected at least 50 rewards in 15 min. Food restriction is then stopped to prepare for surgery. After one week of post-surgery recovery, food restriction is resumed, and rats are put back on the linear track until they perform at least 70 crossings in 15 min. Next, rats are trained on a continuous spatial alternation task in a three-arm radial maze following a single-day acquisition procedure that consist of eight 15 min learning sessions. Training on the alternation task is repeated on 3 days (separated by a rest day), and each day a different arm is chosen as the home arm. On the rest days, the rats perform two 15 min sessions with the same home arm as the day before. Finally, rats are introduced to the sequence memory paradigm as described above until their sequence performance is above 80% by the end of the session, for 3 days in a row. We found that pretraining rats on the alternation task improved the acquisition of the sequence task.

## Surgical procedure

A custom-designed 3D-printed micro-drive array (*Kloosterman et al., 2009*), carrying up to eight tetrodes (four twisted 0.012 mm polyimide- insulated nickel-chrome wires, angled cut; Sandvik, Kista, Sweden) and three stimulation electrodes (two twisted 0.06 mm polyimide-insulated stainless-steel wires, angled cut; California Fine Wire, Grover Beach, CA), was surgically attached to the rat skull

using standard aseptic techniques. To induce anesthesia, the rat was placed in an induction chamber filled with oxygen (0.5–1 L/m) and 5% isoflurane. Next, its head was securely mounted in a stereotaxic frame after shaving the head, and eyes were protected from drying out and prolonged light exposure with eye ointment and aluminum foil. During the surgery, anesthesia was maintained by administration of 0.5–2% isoflurane through a nose mask and adjusted if necessary, based on vital signs: blood oxygen level, heart rate and breathing rate. The body temperature (measured with a rectal probe) was kept constant using a heating pad. After disinfection of the skin with iodine and ethanol, an incision with a scalpel along the mid-line exposed the skull. Small scratches were carved in the bone plates to allow better adhesion of the dental cement used to fix the implant to the skull. Nine anchoring bone screws were inserted (three frontal, two left parietal, two occipital, and one right parietal) and gaps between the screws and the skull were filled up with bio compatible glue (VetBond) and after extra disinfection (10 min Baytril submersion) all screws were fixed together with dental cement. Next, two craniotomies were drilled for rats for the behavioral experiments, and the dura was removed to allow access to the brain above the HC and ventral hippocampal commissure (VHC) (HC-craniotomy center coordinates: 4 mm posterior to Bregma, 2.5 mm right from the midline; VHC-craniotomy center coordinates: 1.3 mm posterior to Bregma, 0.9 mm right from the midline). For bilateral recordings two extra craniotomies were drilled above the HC and VHC on the other hemisphere (HC-craniotomy center coordinates: 4 mm posterior to Bregma, 2.5 mm left from the midline; VHC-craniotomy center coordinates: 1.3 mm posterior to Bregma, 0.9 mm left from the midline). Before mounting of the implant, craniotomies were covered in silicon grease to seal the gaps between the edge of the craniotomies and the tetrode-carrying tubes. Additionally, mineral oil was applied to the tip of the cannulas prior to implantation which prevents back-filling (and possible clogging) of the tetrode-carrying tubes with cerebrospinal fluid or blood. The implant was fixed to the skull with light-curable dental cement (SDI, Bayswater, Australia) and one screw was wired to the electrode interface board to serve as electrical ground. The skin was sutured in the front and back of the implant with surgical threads (4–0 Silk wax coated braided silk, Sofsilk). While the rat was still under light anesthesia, all tetrodes and stimulation wires were lowered 1 mm into the cortex. Finally, 0.7 ml of saline (anti-dehydration) and 0.3 ml Metacam (anti-inflammatory and pain relieve) were administered subcutaneously. The Metacam injection was repeated in the three days following completion of the surgery.

## Electrophysiological recordings

After approximately one week of post-surgical recovery, four to five tetrodes were positioned in the CA1 pyramidal cell layer of the dorsal HC over the course of 3 days to minimize tissue damage. One tetrode was lowered in the white matter above the CA1 cell layer to serve as a reference and one tetrode was placed in the cortex above (parietal association cortex) to aid the online ripple detection. Wide-band (0.1–4 kHz) signals were sampled at 4 kHz, digitized using a 128- channel data acquisition system (Digilynx SX acquisition system with HS-36 analog headstage and Cheetah software; Neuralynx, Bozeman, MO) and saved to a hard disk for offline analysis.

## Online ripple detection and disruption

A live network stream of digitized multi-channel samples from the Digilynx acquisition system was fed into a quad-core workstation that runs the real-time detection software Falcon (*Ciliberti and Kloosterman, 2017*). Falcon initiates TTL pulses via a microcontroller board (Arduino UNO) that is connected to a constant-current stimulator (MultiChannel System, Reutlingen, Germany) which generates a biphasic current for electrical stimulation of the VHC. Here, raw signals of 2–4 electrodes (with the clearest ripples) are first filtered in the ripple band (135–255 Hz) using a Chebyshev type-II IIR filter (order 20). Using the neuralynx acquisition system, the total round trip latency was below 1ms (*Ciliberti and Kloosterman, 2017*). After summing of the ripple power (*RP*) of the different electrodes, ripples were identified based on the following criteria.

$$RP - \mu\left(t\right) > f \cdot mad\left(t\right)$$

In this equation, μ(*t*) and *mad(t)* are the running estimates of the mean and mean absolute deviation calculated on the ripple power, and *f* is a multiplier set to a value in the range 5–13. The value of *f* was adjusted for each daily session to maximize the ratio of positive over false ripple detections. The *RP* is the root mean square of the filtered signal. Estimates of μ(*t*) and *mad(t)* were computed using

an exponential moving average filter with span set to 7 s. Estimates were not updated during a 50ms window after each detection to avoid multiple detections of the same event. To avoid false detections due to heavy head movements or chewing artifacts, the same ripple detection procedure was applied to signals from the cortical electrode. Detections were marked as false positive if a cortical detection occurred within a fixed time window ([–40, 1.5] ms) around a HC detection. While ripple-like events have been described in the parietal association cortex during sleep, only a small fraction is coupled to HC ripples and they have a much lower power compared to HC ripples (*Khodagholy et al., 2017*; *Aleman-Zapata et al., 2022*). In our experiments in the awake state, the limited cortical detections predominantly represent artifacts rather than joint hippocampal-cortical ripples.

The detected hippocampal ripples triggered the generation of a second TTL pulse by Arduino UNO either immediately (disruption condition) or after a random delay between 150 and 250ms. This TTL-pulse was sent to a constant-current stimulator (MultiChannel System, Reutlingen, Germany) to electrically stimulate the VHC. Both the detection event and the time of stimulation (=stimulation event) were sent to the Neuralynx acquisition system for logging. The amplitude of the biphasic electrical pulses (0.2ms duration) varied from 50 to 250 μA and was set in each session to the lowest amplitude that resulted in consistent disruption of hippocampal ripple events. To avoid over-stimulation that could lead to damage of the surrounding tissue, no stimulations could occur within a 150ms lockout period after a prior stimulation. In the disruption condition, this is achieved by discarding detections in a 150ms window after each detection. In the stimulated control condition, all detections that would trigger a stimulation that falls withing a 150ms window after the delayed stimulation are were discarded. Ripple detection and disruption started after rats acquired the task rules reached according to the learning criteria specific for the behavioral task. In the NMTS paradigm, ripple detection and/or disruption occurs only during a trial (i.e. both during and in between the instruction and test phase), but not in the inter-trial time. For every trial one of the control/disruption conditions was picked pseudo-randomly. In the MTS paradigm, ripple detection and/or disruption was applied during both the trials and the inter-trial times intervals. In the sequence memory paradigm, ripple detection and/or disruption was applied during both run epochs and rest epoch. In both the MTS and sequence memory paradigm, the stimulation condition was constant through the whole session.

## Quantification and statistical analysis

Analysis of neural and behavioral data was performed using Python and its scientific extension modules (*Millman and Aivazis, 2011*), augmented with custom Python and C++toolboxes. Throughout the result section, all description summary statistics report mean [99% CI], unless stated otherwise.

## Behavior

Position and speed of the rat were tracked using LED lights mounted on one of the headstages and an overhead video camera (25 Hz). Entry into a maze arm was detected as soon as the animal moved past the first 20 cm of the arm (regardless of the rat actually reaching the reward platform). To assess stereotypical behavior in the MTS task, we defined for every session a stereotypy index. First, a string representing the sequence of arm visits in a trial was created for each of the last 10 trials in a session (for example, '1345' or '5471'). The stereotypy index was defined as the average similarity (based on the normalized Levenshtein distance) between the strings of all trial pairs. The Levenshtein distance measures the minimum number of single character edits that are necessary to change one string into another, and for two strings a and b with lengths |a| and |b| is computed according to the following formula:

$$
\mathrm{lev}\,(a,b) = \begin{cases} |a| & \text{if } |b| = 0, \\ |b| & \text{if } |a| = 0, \\ \mathrm{lev}\,(\mathrm{tail}\,(a)\,,\mathrm{tail}\,(b)) & \text{if } a\,[0] = b\,[0] \\ 1 + \min \begin{cases} \mathrm{lev}\,(\mathrm{tail}\,(a)\,,b) \\ \mathrm{lev}\,(a,\mathrm{tail}\,(b)) \\ \mathrm{lev}\,(\mathrm{tail}\,(a)\,,\mathrm{tail}\,(b)) \end{cases} & \text{otherwise,} \end{cases}
$$

with tail $(x)$ equal to all of string x except for the first character. The similarity is then computed as one minus the normalized Levenshtein distance:

$$\text{similarity}\,(a, b) = 1 - \frac{\text{lev}\,(a, b)}{\max\,(|a|, |b|)}$$

The shuffled distributions used to compute the z-scores for the circularity and stereotypy quantifications of every MTS session, were created by randomizing the visit order of each trial 2000 times.

For behavioral analysis of the sequence paradigm, only sessions in which the rat performed at least 150 visits in the first run epoch are considered. After this selection, data from run 1 and run 2 are merged and we only consider the first 400 visits to assess the (learning) behavior. To compute smoothed learning curves we use a Gaussian kernel with a bandwidth of 50 trials. Next, we calculated the corresponding binomial probability that the (smoothed) number of correct responses is significantly higher (alpha = 0.01) than would be expected by chance (0.5) and define the last trial at which this criteria is met. If the criteria were never met, no learning visit was defined.

## Offline ripple detection

Ripple events were detected in the hippocampal field potential for all trials and sessions of the non-stimulated and stimulated control conditions. For the stimulated control condition, the stimulation artifact is first removed and replaced by cubic interpolation of the signal in a 10ms window around the stimulation. To detect ripples offline, the local field potential recorded from 1 to 3 tetrodes was filtered in the ripple frequency band (140–225 Hz). The ripple envelope was computed as the absolute value of the Hilbert-transformed ripple signal, averaged across the recording sites and smoothed with a Gaussian kernel (bandwidth 15ms). Finally, ripple events were detected when the ripple envelope exceeded a high threshold of $\text{med} + 9 \cdot (\text{Q3} - \text{med})$. Here, med and Q3 represent the median and upper quartile range of the envelope. To determine the start and end time of each ripple, a second threshold was computed with the same method as before, but now only using the signal statistics of a 30ms window before each detected ripple (–100ms to –70ms) and a multiplier of 0.4 instead of 9. Ripple events that were separated by less than 20ms were merged into one, and events with a duration shorter than 40ms were excluded. Offline ripple detection was performed on the same task epochs when online ripple detection was performed. It should be noted that the multiplier for the percentile-based threshold, that we use, should not be compared directly to the multiplier for z-score based threshold that is used in previous studies because of the different statistics being used (median / upper quartile range vs mean / standard deviation). When computing the equivalent z-score multiplier that produces the same absolute threshold as our percentile multiplier, we find that these multiplier values are comparable to the 3–6 standard deviations reported in previous papers (*Girardeau et al., 2009*; *Jadhav et al., 2012*; *Fernández-Ruiz et al., 2019*; *Figure 4—figure supplement 2d e*).

## Evaluation of online ripple detection and disruption

Online ripple detection accuracy was verified for all trials and sessions of the non-stimulated and stimulated control conditions. Online detected ripples were compared to offline detected ripples to identify the fraction of offline ripples that were also detected online (true positive rate or TPR) and the fraction of online detections that did not correspond to an offline detected ripples (false discovery rate, FDR). To assess the disruption of ripples after a closed-loop stimulation of the VHC, the mean ripple envelope after each detection/stimulation was computed in a 30ms time window (from +20ms to +50ms) and normalized to the mean ripple envelope in a 30ms time window (from –50ms to –20ms) before detection/stimulation (=normalized post-detection ripple power). The ripple envelope was computed as described above, including the removal of the stimulation artifact for the disruption session. The artifact removal was also performed at the time of detection for the stimulated control sessions to have a fair comparison between the disruption and stimulated control sessions. The normalized post-detection MUA power was obtained in the same way but using the LFP filtered in 300–2000 Hz range.

## Statistics

To test a difference in means between two (unpaired) samples, we used the Mann-Whitney test. To test the difference in means between three (unpaired) samples we used the Kruskal-Wallis test.

## Acknowledgements

We thank Jyh-Jang Sun, Frédéric Michon and Ta-Shun Su for help with the development of the sequence paradigm and Marine Chaput for setting up the automated maze setup and live video tracking. LD is funded by Research Foundation Flanders (FWO), Belgium as a PhD Fellow fundamental research, grant number 11D9322N. FK is funded by Research Foundation Flanders (FWO), Belgium under grant number G077321N and KU Leuven, Belgium C1 grant C14/17/042

## Additional information

### Funding

| Funder | Grant reference number | Author |
|---|---|---|
| Fonds Wetenschappelijk Onderzoek | PhD fellowship 11D9322N | Lies Deceuninck |
| Fonds Wetenschappelijk Onderzoek | project grant G077321N | Fabian Kloosterman |
| KU Leuven | grant C14/17/042 | Fabian Kloosterman |

The funders had no role in study design, data collection and interpretation, or the decision to submit the work for publication.

### Author contributions

Lies Deceuninck, Conceptualization, Formal analysis, Funding acquisition, Investigation, Visualization, Methodology, Writing – original draft, Writing – review and editing; Fabian Kloosterman, Conceptualization, Software, Supervision, Funding acquisition, Methodology, Writing – review and editing

### Author ORCIDs

Lies Deceuninck http://orcid.org/0000-0002-4695-8551
Fabian Kloosterman http://orcid.org/0000-0001-6680-9660

### Ethics

All experiments were carried out in accordance with protocols approved by KU Leuven animal ethics committee (P119/2015 and P175/2020) and in accordance with the European Council Directive, 2010/63/EU.

### Decision letter and Author response

Decision letter https://doi.org/10.7554/eLife.84004.sa1
Author response https://doi.org/10.7554/eLife.84004.sa2

## Additional files

### Supplementary files

• MDAR checklist

### Data availability

Falcon software for closed-loop ripple detection and code for analysis are publicly available at http://www.bitbucket.org/kloostermannerflab. Source data are deposited in the following Figshare repository: https://figshare.com/s/4c0fcdad7e4890d7ba93.

The following dataset was generated:

| Author(s) | Year | Dataset title | Dataset URL | Database and Identifier |
|---|---|---|---|---|
| Deceuninck L, Kloosterman F | 2024 | Data and software supporting paper 'Disruption of awake sharp-wave ripples have no effect on the immediate behavior nor create short-lasting memories' | https://doi.org/10.25452/figshare.plus.c.6835236 | Figshare, 10.25452/figshare.plus.c.6835236 |

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
