## [Editor Report]

This study reports lack of effect of closed-loop disruption of awake sharp-wave ripples in the repeated-acquisition of spatial memory tasks. These negative results have important theoretical and practical implications in the field of learning and memory. The strength of evidence is solid with methods, data, and analyses broadly supporting the claims.

---

## [Decision Letter]

**Decision letter after peer review:**

Thank you for submitting your article "Awake hippocampal replay is not required for short-term memory" for consideration by *eLife*. Your article has been reviewed by 3 peer reviewers, including Liset M de la Prida as the Reviewing Editor and Reviewer #1, and the evaluation has been overseen by Laura Colgin as the Senior Editor.

Essential revisions (for the authors):

This paper describes the lack of effect of closed-loop SWR disruption in rats experiencing a short-term memory (STM) task. There was an overall agreement that evidence is solid in validating the methodology and controlling for the detection strategy. While we all agree there is potential in the paper, we feel more data and extensive revision is required before a positive recommendation can be made.

In particular:

1) A major point is to provide evidence that the intervention strategy actually works under some conditions. Since this paper reports a lack of effect, we feel that a positive control where SWR disruption results in behavioral effects are necessary. This should come ideally using the same detection protocol and intervention strategies that fail to deliver effects in the tasks reported thus excluding any potential confounding (e.g. bilateral/unilateral disruption, SWR content, SWR replay, long/short SWR, SWR homeostatic effects, filtered ranges for SWR detection., etc….). We understand that reporting negative results is challenging, so we are open to discussing the best strategy to address this point.

2) There are also concerns about whether the type of cognitive function required to solve the task is actually dependent on the hippocampus or not, and/or whether it requires SWR at all. We wonder whether the lack of effect is actually a consequence of the behavioral design or the phase in learning is being used (e.g. behaviour has become already "automatic" by the time of intervention).

3) Given results are in striking contrast with previous data, there is a need for improved discussion and definitions. In case you can provide solid evidence to address major points 1 and 2 above, we would like to see a thorough evaluation of the existing literature and potential similarities/differences that can explain disagreement.

You need to address the comments by the three reviewers having these main issues in mind.

*Reviewer #1 (Recommendations for the authors):*

The main comments are:

- The main conclusion from the experiments is that awake ripples are not required for short-term memory. One issue though is whether there is actually anything in the task that requires using SWRs for recall. The authors may need to show/discuss correlative evidence supporting the hypothesis that ripples are important/non-important in the tasks (i.e., ripple rate is elevated during the key period). Ideally, this should come from their own data.

- Most of the previous reports on SWR intervention have seen effects during the learning phase of the task, but not necessarily once the task rules are learned. Thus, the STM update of task rules may not require SWRs, but learning consolidation does. Can the authors provide/discuss evidence along these lines?

- Since results consist of a lack of evidence there is a need for some positive control. Some questions may arise regarding the effectiveness of the approach due to the lack of effect (e.g are SWRs bilaterally affected?). Can the authors provide one positive experiment where their SWR intervention approach works?

*Reviewer #2 (Recommendations for the authors):*

1) Showing results from a behavioral paradigm affected by the SWR detection and disruption method used here would add a valuable control experiment and give confidence in the approach. How do you know that your SWR disruption affects the SWR content enough, especially if you have to consider a disruption delay of, on average, about 30 ms or 50% of the SWR?

2) The authors should show the SWR duration and amplitude distribution for all three trial types (no stimulation, delayed stimulation, and stimulation). In addition, the authors should show SWR-associated firing rate as a function of time. This could provide insights into the extent to which the SWR content is disrupted.

*Reviewer #3 (Recommendations for the authors):*

Evaluate the possibility of adding a hippocampal place field analysis that serves as a positive control to observe the effects of your ripple disruption implementation in the destabilization of place fields.

---

## [Author Response]

Essential revisions (for the authors):Reviewer #1 (Recommendations for the authors):The main comments are:- The main conclusion from the experiments is that awake ripples are not required for short-term memory. One issue though is whether there is actually anything in the task that requires using SWRs for recall. The authors may need to show/discuss correlative evidence supporting the hypothesis that ripples are important/non-important in the tasks (i.e., ripple rate is elevated during the key period). Ideally, this should come from their own data.

We thank the reviewer for their comment. We show in our data that for all three tasks, more than 80% of all ripples occur either at the reward platforms or the choice point (Figure 1h, Figure 2h, and Figure 3i). At these locations, the ripples and associated replay may serve to mark in memory the path to a collected reward or to evaluate possible future paths. Indeed, previous studies have shown that replay of hippocampal spike sequences at reward site and choice platform appear to be related to future or past paths (Singer et al. 2013; Xu et al. 2019; H. Freyja Ólafsdóttir, Carpenter, and Barry 2017; Tang and Jadhav 2018; Shin, Tang, and Jadhav 2019; Foster and Wilson 2006; Karlsson and Frank 2009; Ambrose, Pfeiffer, and Foster 2016; Gupta et al. 2010; H. Freyja Ólafsdóttir et al. 2015; Carey, Tanaka, and Meer 2019). In particular, Xu et al. 2019 looked at replay in a working and reference memory task on the 8-arm radial maze, which are very similar to the tasks that we chose to use, and observed that correct choices by the rats are preceded by replay representing that correct choice. We have adapted the text in the introduction and discussion to more clearly make this point.

- Most of the previous reports on SWR intervention have seen effects during the learning phase of the task, but not necessarily once the task rules are learned. Thus, the STM update of task rules may not require SWRs, but learning consolidation does. Can the authors provide/discuss evidence along these lines?

The reviewer is correct that most of the SWR interventions have been performed during the learning phase of a novel task (Fernández-Ruiz et al. 2019, Jadhav et al. 2012; Igata, Ikegaya, and Sasaki 2021). The acquisition of a novel task involves a number of cognitive processes, including short- and long-term memory, building a map of the environment, exploration of the solution space and incorporating (non-)rewarding feedback. Based on available evidence, SWRs could contribute to many of these processes. Our experiments were designed to isolate the short-term updating of known task rules and assess the contribution of SWRs. In our current manuscript, we refer to this as the ‘immediate and intermediate timeframe of impact’. The data is more consistent with the reviewer’s model that (quick) update of task information does not require SWRs, but rather that SWR mediate more gradual updates and strengthening of representations (i.e., consolidation over a longer timeframe). We have expanded our discussion to more clearly make these points (lines 407-424 and lines 456-472).

- Since results consist of a lack of evidence there is a need for some positive control. Some questions may arise regarding the effectiveness of the approach due to the lack of effect (e.g are SWRs bilaterally affected?). Can the authors provide one positive experiment where their SWR intervention approach works?

In consultation with the editors, we have addressed this comment in two ways.

First, we have performed additional experiments (n=3 rats) with bilateral hippocampal recordings that show that our unilateral approach for stimulating the ventral hippocampal commissure disrupts hippocampal ripple activity in both hemispheres. These data show that our manipulation has a broad effect on hippocampal activity and that the intervention approach works.

Second, we have previously shown that the same hippocampal ripple disruption approach during a 2h rest period after learning results in a behavioral deficit (Michon et al., 2019), pointing to a role for SWRs in consolidation. Although we did cite this paper in the Discussion, we have now also added this reference to the Results section when describing the ripple detection and disruption method and performance (lines 120 and 347).

Reviewer #2 (Recommendations for the authors):1) Showing results from a behavioral paradigm affected by the SWR detection and disruption method used here would add a valuable control experiment and give confidence in the approach. How do you know that your SWR disruption affects the SWR content enough, especially if you have to consider a disruption delay of, on average, about 30 ms or 50% of the SWR?

In consultation with the editors, we have addressed the comment about a positive control experiment in two ways.

First, we have performed additional experiments (n=3 rats) with bilateral hippocampal recordings that show that our unilateral approach h for stimulating the ventral hippocampal commissure disrupts hippocampal ripple activity in both hemispheres. These data show that our manipulation has a broad effect on hippocampal activity and that the intervention approach works.

Second, we have previously shown that the same hippocampal ripple disruption approach during a 2h rest period after learning results in a behavioral deficit (Michon et al., 2019), pointing to a role for SWRs in consolidation. Although we did cite this paper in the Discussion, we have now added this reference to the Results section when describing the ripple detection and disruption method and performance (lines 120 and 347).

The reviewer makes a valid point about the disruption delay and the concern that incomplete disruption of activity at the beginning of SWRs may explain the lack of behavior effect. It is important to note that the disruption delay is inherent to closed-loop SWR disruption approaches used to date and previous studies that disrupted SWRs still reported behavioral performance deficits in range of 10-20% (Michon et al. 2019, Girardeau et al. 2009, Jadhav et al. 2012, Fernández-Ruiz et al. 2019). Unfortunately, not all papers report statistics about SWR detection delay. However, given that the detection algorithms used in these papers is very similar, it is also expected that the detection delays are comparable. The detection delays that we measured in the current study are similar to what we reported before in a study that did show behavioral deficits (Michon et al., 2019). Thus, we are confident that even with the reported disruption delay it should have been possible to observe an effect on behavioral performance. Still, we cannot exclude that the initial part of the SWR is more important for the kind of tasks that we tested and we now elaborate on this limitation in the Discussion (lines 480-488).

2) The authors should show the SWR duration and amplitude distribution for all three trial types (no stimulation, delayed stimulation, and stimulation). In addition, the authors should show SWR-associated firing rate as a function of time. This could provide insights into the extent to which the SWR content is disrupted.

We thank the reviewer for this suggestion. We added to the manuscript the ripple peak amplitude distribution for the no stim. and stim. control conditions (Figure 4 panel c). Because in the disruption condition the ripples are disrupted we cannot quantify the relative delay nor the peak amplitude of the ripple.

For the behavioral experiments we have only recorded LFP and have thus no access to the SWRassociated firing rates of the hippocampal cells. In our extra experiments, which use the exact same technique for recording and disruption we do have access to spiking data and added it to Figure 4 —figure supplement 2. Here, we also quantified the normalized post-detection MUA power in the same way as the normalized post-detection ripple power. This shows that the SWRassociated firing rate drops significantly when a stimulation occurs right after the detection of a SWR (e.g., in disruption sessions).

Reviewer #3 (Recommendations for the authors):Evaluate the possibility of adding a hippocampal place field analysis that serves as a positive control to observe the effects of your ripple disruption implementation in the destabilization of place fields.

Unfortunately we have no unit data available in our recordings to perform place field analysis.